# Antibacterial Activity of Solvothermal Obtained ZnO Nanoparticles with Different Morphology and Photocatalytic Activity against a Dye Mixture: Methylene Blue, Rhodamine B and Methyl Orange

**DOI:** 10.3390/ijms24065677

**Published:** 2023-03-16

**Authors:** Ludmila Motelica, Ovidiu-Cristian Oprea, Bogdan-Stefan Vasile, Anton Ficai, Denisa Ficai, Ecaterina Andronescu, Alina Maria Holban

**Affiliations:** 1National Research Center for Micro and Nanomaterials, University Politehnica of Bucharest, 060042 Bucharest, Romania; 2National Research Center for Food Safety, University Politehnica of Bucharest, Splaiul Independentei 313, 060042 Bucharest, Romania; 3Faculty of Chemical Engineering and Biotechnologies, University Politehnica of Bucharest, 1-7 Polizu St., 011061 Bucharest, Romania; 4Academy of Romanian Scientists, Ilfov Street 3, 050044 Bucharest, Romania; 5Microbiology & Immunology Department, Faculty of Biology, University of Bucharest, 077206 Bucharest, Romania

**Keywords:** ZnO, antibacterial, methyl orange, methylene blue, rhodamine B, photocatalysis, solvothermal

## Abstract

In this paper, we report the synthesis of ZnO nanoparticles (NPs) by forced solvolysis of Zn(CH_3_COO)_2_·2H_2_O in alcohols with a different number of –OH groups. We study the influence of alcohol type (n-butanol, ethylene glycol and glycerin) on the size, morphology, and properties of the obtained ZnO NPs. The smallest polyhedral ZnO NPs (<30 nm) were obtained in n-butanol, while in ethylene glycol the NPs measured on average 44 nm and were rounded. Polycrystalline particles of 120 nm were obtained in glycerin only after water refluxing. In addition, here, we report the photocatalytic activity, against a dye mixture, of three model pollutants: methyl orange (MO), methylene blue (MB), and rhodamine B (RhB), a model closer to real situations where water is polluted with many chemicals. All samples exhibited good photocatalytic activity against the dye mixture, with degradation efficiency reaching 99.99%. The sample with smallest nanoparticles maintained a high efficiency >90%, over five catalytic cycles. Antibacterial tests were conducted against Gram-negative strains *Salmonella enterica* serovar Typhimurium, *Pseudomonas aeruginosa,* and *Escherichia coli*, and Gram-positive strains *Enterococcus faecalis*, *Bacillus subtilis*, *Staphylococcus aureus*, and *Bacillus cereus*. The ZnO samples presented strong inhibition of planktonic growth for all tested strains, indicating that they can be used for antibacterial applications, such as water purification.

## 1. Introduction

Water is one of the most precious resources but at the same time clean sources become scarcer as human negligence and industrial activities lead to high pollution levels [1]. The organic substances represent the most important class of pollutants, from dyes to antibiotics or pesticides [2]. In addition, microbial contamination represents a health problem, leading to many deaths worldwide from dysentery, typhoid fever, and cholera [3].

The traditional methods for water purification include processes, e.g., sedimentation [4], flotation [5], filtration [6], adsorption [7], or chlorination [8], but the associated costs are important and they have limitations (e.g., do not remove the antibiotics). The use of nanomaterials for water purification is still in its infancy. The most promising nanomaterials are those that exhibit photocatalytic and antimicrobial activities [9]. Such nanomaterials could achieve two processes, removing the organic pollutants and killing pathogenic microorganisms, under visible light irradiation, in a single step.

Zinc oxide (ZnO) is a highly versatile material, with strong photocatalytic activity, when exposed to ultra-violet (UV) light, similar to TiO_2_. Nevertheless, due to its lower band-gap energy, ZnO can act as a photocatalyst also under visible light irradiation. ZnO is also an antimicrobial material, with increasing activity as the dimension of nanoparticles (NPs) decrease. In fact, the antimicrobial mechanisms employed by the ZnO nanoparticles can be classified as light related and morphology-related [10]. When irradiated, the ZnO NPs produce reactive oxygen species (ROS) responsible for both photocatalytic and antimicrobial activities. Under dark conditions, the ZnO still exhibit antimicrobial activity due to mechanical damage of the cell’s membranes (puncture, penetration, rupture etc.), and probably due to release of Zn^2+^ ions that can interact with essential constituents of the cells [11]. In addition, the Food and Drug Administration (FDA) consider ZnO as generally recognized as safe (GRAS), which has a direct impact on the potential uses: cosmetics, sunscreens, food packaging, paints, medical structures, drug delivery, dentistry, wound dressing, textile industry and other antimicrobial applications [12,13,14,15]. Therefore, ZnO NPs can be considered as a potential ideal candidate for water treatment processes. It is considered not toxic for humans, does not dissolve in water, and can be easily separated and reused, it has a high photocatalytic activity against single dye model solution, and finally, it has antimicrobial activity. Nevertheless, these properties depend on the size, shape and surface morphology, and defects of ZnO NPs, all of which depend on the synthesis method.

The literature abounds with various methods for ZnO NPs synthesis: spray pyrolysis, precipitation, chemical vapor deposition, thermal decomposition, natural extracts, forced hydrolysis or solvolysis, ionic liquids, sol-gel or microwave [16,17,18]. Forced solvolysis in alcohols is seldom reported, and studies still need to be carried out. In our recent study [19], we report that ZnO is obtained in various primary alcohols, from methanol to 1-hexanol. The morphologies change from round, to polyhedral, to nanorods as the number of carbon atoms of the alcohol increases. At the same time, the syntheses in secondary or tertiary alcohols proved to be inefficient. In another study ZnO nanorods were obtained in primary alcohols, at high temperatures of 250–300 °C in an autoclave [20]. Unfortunately, because only a 2 h reaction time was employed, the final product was impurified with zinc acetate. Increasing the reaction time to 4 h, at 170 °C, spherical and rod-like nanoparticles were obtained [21]. The computer aided modeling indicated the probability that the smallest crystallite would be obtained in n-butanol. By employing forced solvolysis, the NPs surface will usually have a larger defect density, which offers active centers for photocatalytic activity and ROS generation.

There are many literature reports about photocatalytic activity of ZnO NPs, but usually on single model pollutant (dye) [22]. The water is seldom polluted with a single species, usually wastewaters containing a mixture of organic compounds, from different classes [23]. This raises the question of whether ZnO NPs can degrade a more complex organic mixture, as competition for photocatalytic sites can occur. In our recent study, the photocatalytic activity was studied against five individual model dyes (eosin Y, methyl orange, gentian violet, rhodamine B and methylene blue) [19]. At the same time, the literature is scarce in reports of photocatalytic activity on dye mixtures [24]. Such study is possible only if the dyes have different light absorption domains so they can be individually monitored from the mixture. This constrains the dye types that can be mixed, an obvious choice being the mixture of methyl orange (MO) with rhodamine B (RhB) and methylene blue (MB). This mixture also has the advantage of using dyes of three different types [25]. Therefore, we report here the obtaining of ZnO by forced solvolysis in different alcohols, and its capacity to degrade a mixture of different dyes (MO + RhB + MB). Coupling the photocatalytic activity with the antibacterial properties of ZnO NPs indicates the possibility that such nanomaterials be used for water treatment.

## 2. Results and Discussion

### 2.1. Thermal Analysis—Thermogravimetry (TG)—Differential Scanning Calorimetry (DSC)

A sample can contain an amorphous part and a crystalline part, the last one being the only one identified by XRD. Therefore, in order to determine the presence of amorphous impurities other suitable analysis must be employed. The white nanopowders obtained by forced solvolysis were firstly investigated by thermal analysis (TG coupled with DSC). The purity of the samples was assessed by TG-DSC, as a complementary tool for powder X-ray diffraction. The expected products beside ZnO can contain Zn(OH)_2_, unreacted Zn(CH_3_COO)_2_·2H_2_O, and various hydroxy-acetate polymeric species, the most important being Zn_5_(CO_3_)_2_(OH)_6_ and Zn_5_(OH)_8_(CH_3_OO)_2_. The results are presented in Figure 1.

For the samples ZnO_B and ZnO_EG the residual mases obtained at 900 °C are 98.47% and 97.60% respectively, indicating that direct synthesis of ZnO in the n-butanol or ethylene glycol was successful. The most important data are presented in Table 1.

The ZnO_B presents a mass loss of 0.41% up to 200 °C, indicating the presence of some –OH moieties and volatile molecules on the nanoparticles surface. Above 200 °C, the sample loses 1.12% of its mass with a slow pace, the process being accompanied by a weak exothermic effect with maximum at 357.8 °C, indicating the oxidation of some trace organic impurities.

In the case of the ZnO_EG sample, the mass loss exhibited in the interval RT-200 °C is similar, 0.47%. The sample is losing 1.92% in a quick process, associated with an endothermic effect at 275.8 °C, indicating a decomposition/desorption of organic impurities.

The sample ZnGly present a single step decomposition between 320–420 °C, with a mass loss of 48.18%, accompanied by an exothermic effect at 386.1 °C. The residual mass at 900 °C consists of ZnO and amounts to 51.14%. This result shows that, in case of glycerin, the direct obtaining of ZnO is not possible. The XRD analysis for this sample indicated that the obtained powder contains zinc glycerolate, JCPDS 023-1975, as shown in Appendix A. Further SEM analysis of ZnGly powder indicates the presence of micrometric rose-like structures (Appendix A) obtained from the agglomeration of polyhedral palettes sprinkled with 200 nm hexagonal particles. Flower like structures are reported previously for zinc glycerolate, but chrysanthemum-like [26].

However, the TG-DSC analysis of ZnO_G sample obtained after boiling the ZnGly in water indicates the successful synthesis of ZnO, as the residual mass is 98.09%. The ZnO_G sample is slowly losing 1.19% of its mass up to 200 °C, indicating a higher concentration of water/hydroxyl moieties on the ZnO nanoparticles surface. After 200 °C the slow mass loss continues up to ~350 °C, amounting to 0.72%, with the process being accompanied by an exothermic effect at 341.5 °C, indicating the oxidation of organic impurities. TG/DSC analyses for individual ZnO_B, ZnO_EG, and ZnO_G samples are presented in Appendix A.

By coupling these results with the XRD analysis, we can state that in case of ZnO_B, ZnO_EG, and ZnO_G samples, the ZnO was successfully obtained.

### 2.2. X-ray Diffraction Analysis (XRD)

Determination of the composition of the crystalline phase was conducted by XRD analysis, and the results are presented in Figure 2. The obtained diffractograms show the pattern for ZnO (JCPDS card no. 80-0075) for all three samples, ZnO_B, ZnO_EG, and ZnO_G, with the Miller indices being assigned for each peak. 

The nano-nature of the samples is revealed by the broadening of the XRD peaks. For the sample ZnO_EG a small shift towards smaller values of 2θ can be noticed. The crystallite average size D, lattice parameters and microstrain ε were calculated by Rietveld refinement (Table 2). The crystallite size might be different from particle dimension and can be considered the size of a coherently diffracting domain [27]. The calculated values indicate an increase of the *c/a* ratio from ZnO_B to ZnO_G and further to ZnO_EG samples.

The positional parameter, *u*, that can be calculated with Equation (1), represents the displacement measure of each atom when comparing to the next on the *c* axis [28]:*u* = (*a*^2^/3*c*^2^) + 0.25(1)

Equation (1) indicates that if the positional parameter *u* increases, then the *c/a* ratio decreases, and this leads to higher microstrain values (ε) [27].

The length of dislocation lines in a unit volume represents the dislocation density (δ), which is calculated according to Equation (2), where D is the crystallite size:δ = 1/D^2^(2)

As the crystallite size decreases, the value of δ increases sharply and practically represents the amount of defects in a sample [29]. A high concentration of surface defects might lead to the creation of active centers in photocatalysis, which will increase the performance of the nanoparticles in the degradation of the organic pollutants under light irradiation. Furthermore, an increased level of reactive oxygen species (ROS) generation will lead to a higher antimicrobial activity for a particular sample.

The calculated values for δ indicate that the sample ZnO_B has the highest concentration of surface defects, while ZnO_G and ZnO_EG exhibit approximately half that value.

### 2.3. Transmission Electron Microscopy (TEM)

In order to investigate the shape and dimension of the nanoparticles, transmission electron microscopy bright-field images were recorded for all three samples (Figure 3, Figure 4 and Figure 5). 

As can be observed from Figure 3, for the sample ZnO_B, the XRD calculated crystallite dimension value is in good agreement with the particles size. The calculated NPs size distribution indicates an average nanoparticle dimension of 26.47 nm (Appendix A). Therefore, each nanoparticle contains a single crystallite, making them monocrystalline. The nanoparticles are polyhedral shaped, with a low tendency to agglomerate. From the selected area diffraction pattern (SAED), as shown in the inset of Figure 3a, we can conclude that the hexagonal ZnO (JCPDS card no. 80-0075) is the only identified crystalline phase. The HRTEM image (Figure 3d) shows clear lattice fringes of interplanar distance d = 0.259 nm. Corresponding to Miller indices (0 0 2). 

For the sample ZnO_EG the NPs have rather a rounded shape with no agglomeration tendency (Figure 4). The size of the nanoparticles is larger than that for the ZnO_B sample, with the NPs size distribution (Appendix A) indicating a value of 44.24 nm, in accordance with XRD calculation. These nanoparticles are also monocrystalline, as their size matches the previous calculated value for the crystallite size. The SAED image (inset of Figure 4a) confirms the presence of crystalline ZnO, while the HRTEM image (Figure 4d) permits the determination of the interplanar distance of 0.259 nm corresponding to Miller indices (0 0 2).

We can conclude that the change of the solvent has an impact on the morphology of the nanoparticles, with the presence of ethylene glycol favoring the round shape, while the n-butanol generates polyhedral shape nanoparticles.

For the ZnO_G sample, due to differences in the synthesis process, the morphology is very different from the previous samples (Figure 5). The particles are large, often over 100 nm, hexagonal, with some agglomeration tendency, and no amorphous phase. The particles size distribution (Appendix A) indicates a value of 120.15 nm. A closer look at the hexagonal particles will reveal that they are composed from many smaller polyhedral crystallites, as shown in Figure 5c. Therefore, these hexagonal particles can be considered polycrystalline. The HRTEM image (Figure 5d) shows clear lattice fringes in a regular succession of the atomic planes, indicating uniformity of the structure.

To summarize, the samples are of different size and morphology. The smallest nanoparticles are ZnO_B at < 30 nm and the largest are ZnO_G at ~120 nm. The ZnO_B sample presents polyhedral monocrystalline nanoparticles, ZnO_EG has rounded monocrystalline nanoparticles, and ZnO_G has hexagonal polycrystalline particles.

### 2.4. Spectroscopic Studies

#### 2.4.1. FTIR Spectroscopy

FTIR spectroscopy was employed to investigate the purity of the obtained ZnO NPs (Figure 6). As expected, the presence of Zn-O bond stretching vibrations was observed to generate the very strong absorption band under 500 cm^−1^, with peaks at 457 and 410 cm^−1^ [30]. The presence of –OH surface moieties is revealed by the weak bands from 614 and 685 cm^−1^ attributed to the Zn-OH bending vibrations [31]. The broad band from 3400 to 3500 cm^−1^ with weak intensity is attributed to the stretching vibration υ_OH_, indicating the presence of adsorbed water or –OH groups.

The double peaks from 1400 to 1600 cm^−1^ interval are attributed to the carboxylate group, 1435 cm^−1^ to the asymmetric ν_as(COO)_, and 1568 cm^−1^ to the symmetric ν_s(COO)_ vibrations, respectively. The carboxylate group is in ionic form as the value of difference between these two peaks is in the 100–200 cm^−1^ interval. This indicates that some acetate ions remain adsorbed in the samples, in concordance with the exothermic effect present on the DSC curve due to the oxidation of these ions. 

The weak double peaks just under 3000 cm^−1^ are assigned to the symmetric and asymmetric –C_sp3_-H vibrations from the alcohol used. The band from 1340 cm^−1^ can be attributed to the symmetric bending (δ_s_CH_3_) vibration [32], while the band from 1033 cm^−1^ can be attributed to the -C-O stretching vibration. Finally, the band from 900 cm^−1^ is due to the presence of tetrahedral Zn^2+^ ions [33].

#### 2.4.2. UV-Vis Spectroscopy

The white ZnO powders reflects the visible light. Therefore, the samples absorbance is small in the 400–900 nm domain. Characteristic for ZnO is the strong absorption of the UV radiation, with a maximum at 366 nm (Figure 7a).

This strong absorption under 400 nm makes ZnO nanoparticles suitable as a protective coating [34,35], as sunscreen in cosmetics [36,37], as well as in the textile industry [38,39,40]. The strong absorption band from the UV domain represent the energy required for the transition of electrons between valence and conduction bands (VB to CB). The band-gap energy value for ZnO is therefore calculated based on this transition. By using Equation (3) of the Kubelka–Munk function F (R), where R is the diffuse reflectance of the sample:F(R) = (1 – R)^2^/2R(3)
the energy values for the direct band-gap can be determined by graphical extrapolation to [F(R)∙hν]^2^ = 0 (Figure 7b).

The theoretical value for the band-gap energy is 3.37 eV and is higher than the calculated band-gap energy values for all three samples. Nevertheless, they are in concordance with other literature reports [41,42]. The existence of crystalline defects on the nanoparticle surface, as determined by the XRD data, creates inside the band-gap additional electronic levels, which in turn leads to smaller energy values [43].

The variation of the band-gap energies (Table 3) is in good agreement with the values calculated for δ (Table 2). The ZnO_B sample with the highest δ value present a high surface defects density and has the lowest band-gap value. The ZnO_EG and ZnO_G samples have similar values for δ, smaller than for ZnO_B, and therefore their band-gap values are similar, but larger than that of the ZnO_B sample.

In conclusion, the defect density has a direct influence on the band-gap energy values [36]. A lower value of the band-gap energy requires less energy to promote the electrons from VB to CB, and therefore they are expected to perform better as photocatalysts.

#### 2.4.3. Photoluminescence Spectroscopy

The fluorescence (photoluminescence) spectra of the ZnO, Figure 8, usually present two emission zones, one in the UV domain, just under 400 nm, and one in the visible region [44,45,46,47]. The emission peak near band edge (NBE) is usually located in UV domain, most often in the interval 380–400 nm, and is the result of exciton recombination [48,49,50]. When the nanoparticles have multiple surface defects that can block the recombination process by trapping the free electrons, the NBE will usually have lower intensity when compared with the visible emission bands [51,52].

The visible emission bands from the fluorescence spectrum of ZnO are directly related to the presence of crystalline defects and are named deep level emission (DLE). In pure samples, a series of defects might exist: zinc vacancies (V_Zn_), zinc interstitials (Zn_i_), oxygen vacancies (V_O_), oxygen interstitials (O_i_), or oxygen anti-sites (O_Zn_) [53,54]. The emission in the violet region is usually assigned to the V_Zn_ while the blue emission (457 nm) is attributed to the transitions from Zn^+^_I_ to valence band (VB). The band from 481 nm is generated by the transitions from Zn^+^_I_ to V_Zn_. The oxygen defects, V_O_ and V_O+_, are responsible for the green emission from 513 nm [55]. The ZnO_B sample presents the strongest emission intensity for both NBE and DLE bands, indicating a higher density of surface defects, while for the ZnO_G sample the NBE band is placed at the lower value of 392 nm and is less intense [56].

### 2.5. Photocatalytic Study

The photocatalytic activity was investigated against a mixture of three dyes: methyl orange, rhodamine B, and methylene blue solution (Figure 9a). The mix was chosen as these dyes are commonly used as organic pollutants models for waters. The MB is a phenothiazine dye, with a monomer-dimer equilibrium, that can be observed by absorption maxima at 664 nm and 614 nm, respectively [57,58,59]. A competition for the catalytic centers between monomer and dimer is possible in the case of MB [60], but also between MB, RhB, and MO. The obtained results (Figure 9b–d) indicate that all maxima are decreasing, so we can conclude that all dyes are degraded by the ZnO NPs simultaneous.

RhB is part of triarylmethane dyes class, with a higher resistance to photo-degradation [61]. It is important to know the susceptibility of organic dyes to the ZnO NPs photo-degradation because there are specific applications where high photocatalytic activity is unwanted. The textile industry is such an example, where ZnO NPs are employed as an antimicrobial agent and as a UV-shield, being embedded in the fibrillary structure [61]. If ZnO NPs exhibit high photocatalytic activity in this case, they would degrade the organic support that they are meant to protect [62,63,64]. Therefore, such behavior is not desirable. 

For low concentration solutions, the photo-degradation reactions follow an apparent first-order kinetics, described by Equation (4):ln (C_0_/C) = *k*_app_∙t(4)
where C is the dye concentration at specific time (C_0_ representing the initial concentration) and *k*_app_ is the rate constant. By plotting ln (C_0_/C) vs. time (Figure 10a–c), the *k*_app_ values can be determined. The R^2^ values, determined *k*_app_ and degradation efficiency, are given in Table 4.

All three ZnO samples have a good degradation efficiency under visible light irradiation, exhibiting strong photocatalytic activity against all three dyes in the mixture. The proposed photocatalytic degradation mechanism implies the generation of ROS and is depicted in Figure 11.

The best degradation efficiency was obtained for ZnO_B sample with 99.99% for MO and RhB dyes, and 99.18% for MB. ZnO_G sample, with large hexagonal particles, still attained a degradation efficiency of 91.07% for MO, but only 74–75% for MB and RhB. These results are in good agreement with the calculated values for δ (XRD) and PL intensities, indicating that the surface defects are generating the ROS capable to degrade the organic compounds, and a higher defect density will lead to better photocatalytic activity. Such defects represent catalytic centers for the photo-degradation of the organic molecules. Surprisingly or not, the dyes in the mixture do not behave as they do in single solutions. In our recent study [19], the MB together with MO were the most susceptible dyes, with a degradation efficiency of 97–98% after 40 min irradiation (the individual solutions were three times more concentrated, without being mixed). RhB attained merely 54.8% degradation. Here, the ZnO shows clear preference for MO and RhB over MB in the mixture, indicating a partially selective photocatalytic activity. As this behavior is common for all three ZnO samples, we have to hypothesize that MO is more susceptible when mixed with the other two cationic dyes. The literature reports similar behaviors where the selectivity is changed by mixing the dyes. The composite ZnO/TiO_2_/F-doped SnO_2_ under UV irradiation presents a degradation efficiency at 40 min of 42%, 81%, and 99% for individual dyes RhB, MO, and MB [65], similar with the values reported by us under visible light [19]. However, in the mixture RhB + MO + MB under the same UV radiation, the degradation efficiency increased to 97–98% for all dyes. By mixing ZnO with graphene oxide (GO), a higher selectivity for MB was reported in the mixtures MO/MB and RhB/MB, probably due to the fact that MB is a cationic dye and GO has available π electrons for interactions [66]. In another study, the ZnO composite with BiOX (X = Cl, Br or I) had a degradation rate MO>MB>RhB for X = Cl and Br, but the degradation efficiency changed to MO>RhB>MB for X = I [67]. In a recent study where individual and mixtures of up to four dyes were degraded under UV and solar radiation, variable degradation efficiency were found for composite graphene oxide/ Sn-doped ZnO catalysts [68]. If only MO and MB dyes are presented in solution, the same degradation efficiency of over 99% is reported under solar light irradiation in 1 h. By mixing MO with MB, RhB and methyl red, the degradation efficiency after 6 h of UV irradiation are changed to 96%, 85%, and 39% for MB, RhB, and MO, indicating again the mutual influence that dyes have on each other’s degradation.

Giving the high degradation efficiency of the ZnO_B sample, we have tested the reusability of the ZnO NPs in five photocatalytic cycles (Figure 10d). The sample retained high photocatalytic activity, with a degradation efficiency of over 90% against all dyes over five cycles. While for MO the efficiency remains over 98%, for RhB, and especially for MB, the decrease is higher, indicating that these molecules are somehow more resistant.

### 2.6. Antimicrobial Assay

To confirm the potential antibacterial activity of the ZnO NPs, the evaluation of the zone of inhibition of growth was firstly employed. The ZnO NPs present a different antibacterial effect, depending on the tested strain and nanoparticles size. We utilized Gram-negative strains (*E. coli*; *S. typhimurium*; *P. aeruginosa*) and -positive strains (*S. aureus*; *B. cereus*; *B. subtilis*; *E. faecalis*), relevant in water-borne infections, in order to highlight the antibacterial spectrum of the obtained ZnO NPs. The obtained results, shown in Figure 12, indicate that the bacterial growth inhibition zone is influenced by the size of the nanoparticles for all the samples.

The highest values of diameter of inhibition of growth were obtained for ZnO_B NPs, with these NPs also exhibiting the smallest nanoparticles size. The ZnO_G sample, with larger nanoparticles and visible agglomerations present the lowest values for inhibition diameters. *P. aeruginosa* and *B. subtilis* can be considered the most resistant strains, with diameters of inhibition of growth of just under 20 mm. *P. aeruginosa*, one of the most resistant pathogens in wound infections, exhibited the smallest diameter of inhibition zone, around 10 mm [69]. The remaining bacterial strains can be considered as being very susceptible to the action of ZnO_B NPs, with *E. faecalis* being the most sensitive, exhibiting a diameter of inhibition surpassing the 30 mm value. The results indicate that small polyhedral ZnO nanoparticles have a strong disinfection potential, even against antibiotic-resistant microorganisms [70]. There are two antimicrobial mechanisms for ZnO nanoparticles. One is based on ROS production under light irradiation, and the second one is related to the size and morphology of the nanoparticles (larger ones being less active) [71,72]. We performed the antibacterial assay with no light irradiation. Therefore, we confirm that the antibacterial activity of ZnO depends on the size and shape of the nanoparticles. The smaller nanoparticles, e.g., the ZnO_B, are more potent, as expected, when compared with the larger ones [71].

Planktonic growth inhibition results showed that small nanoparticles are more active. Zn^2+^ ions could be released from the ZnO NPs, since they affect the development of free-floating cells. Moreover, smaller ZnO NPs remain in suspension and can interact with the cells, adhere to the membrane, or penetrate it. Figure 13 reveals that the highest bacterial growth inhibition in nutritive broth is obtained for the ZnO_B sample with the smallest nanoparticles. However, important growth inhibition was also obtained for the other samples, indicating the importance of nanoparticles size, but also the existence of additional inhibition mechanisms, e.g., the release of zinc ions. Nevertheless, the samples ZnO_EG and ZnO_G exhibited lower antibacterial activities, with OD >0.1, with the results in the aforementioned situations showing only a partial inhibition of bacterial growth rather than a whole eradication.

Pathogenic bacteria represent a critical threat for health worldwide, and the resistance to the antibiotics is becoming a serious problem. Alternative fighting methods include the use of non-traditional antimicrobials, such as nanoparticles or plant extracts. ZnO can be considered as a nano-antibiotic, with a strong action against pathogenic microorganisms. The antimicrobial mechanisms reported by the literature are based on ROS production under UV or visible light irradiation, but are also dependent on the size and morphology of the nanoparticles [71,72]. Figure 14 depicts the multiple pathways described by the literature: ROS production, direct damage inflicted by nanoparticles to the microorganisms’ membrane or release of Zn^2+^ ions that can interact with cellular constituents [73,74,75].

The nanoparticles can physically damage the bacterial membrane by puncture, penetration and internalization, provoking leakage of cytoplasm, leading to the cells’ death [74]. In addition, once internalized, the ZnO nanoparticles can generate ROS inside the cell or can release Zn^2+^ ions that will damage the DNA, mitochondria, proteins, or other components that are vital for the surviving of the bacteria. When nanoparticles are adherent to the bacterial membrane, the ROS species will be generated at the interface between ZnO nanoparticles and the cellular wall. ROS production will take place, with higher intensity under light irradiation, and remains the main bactericidal mechanism [73]. The plasma membrane is damaged by the oxidative stress generated by the accumulation of the ROS. Smaller ZnO nanoparticles, having a larger surface, in general produce more ROS, and therefore exhibit a stronger antibacterial activity [76]. As previously reported, smaller nanoparticles tend to cluster near bacterial cells, and this leads to higher bactericidal activity, due to ROS being produced next to the cell wall [19].

This study indicates the possibility to use ZnO NPs for water disinfection, with the ZnO_B sample exhibiting a broad antibacterial activity against all tested strains, both Gram-negative and Gram-positive. Considering the ROS generation under light irradiation, a more potent microbicide effect can be obtained.

## 3. Materials and Methods

Zinc acetate was obtained from Merck (Merck Group, Darmstadt, Germany). The alcohols, n-butanol (B), ethylene glycol (EG), and glycerin (G) were used as received from Sigma (Redox Lab Supplies Com SRL, Bucharest, Romania), without further purification.

For each typical ZnO NPs synthesis, we used 50 mL of each alcohol, in which 5 g of Zn(CH_3_COO)_2_·2H_2_O were added under magnetic stirring, and the solution was kept overnight at 80 °C. The heating was removed after 24 h and the solution were allowed to rest another 24 h at 25 °C. The white powder was separated, centrifuged, and washed with absolute ethanol thrice, followed by drying in an electrical oven at 80 °C. The obtained ZnO nanopowders were named in correspondence with the alcohol used as solvent (Table 5):

As the white powder obtained from glycerin synthesis was proven to be zinc glycerolate, it was further placed with 50 mL distilled water in a round flask under magnetic stirring at reflux for additional 24 h. The precipitate was again separated by centrifugation and washed with absolute ethanol thrice, followed by drying in electrical oven at 80 °C, resulting the ZnO_G sample that contained zinc oxide.

The thermal analysis was performed using a Netzsch STA 449C Jupiter (Selb, Germany). Dry powder samples of ~20 mg were placed in an open Al_2_O_3_ crucible and heated up to 900 °C with a 10 °C∙min^−1^ rate, under a constant flow of dried air (50 mL∙min^−1^). An empty alumina crucible was used as a reference.

The Fourier transform infrared spectra (FTIR) were recorded with a Nicolet iS50R spectrometer (Thermo Fisher Scientific, MA, USA), using the attenuated total reflection (ATR) accessory. Each spectrum represents the average of 32 scans made at a resolution of 4 cm^−1^_,_ in the interval 400 and 4000 cm^−1^.

The UV-Vis spectra were recorded with a JASCO (Easton, PA, USA) V560 spectrophotometer equipped with a 60 mm integrating sphere (ISV-469) in the diffuse reflectance mode. All the measurements were made with a speed of 200 nm min^−1^, in the domain 200–900 nm.

The fluorescence (PL) spectra were recorded with a Perkin Elmer P55 (Perkin Elmer, Waltham, MA, USA) fluorimeter. A Xe lamp was used as an excitation source at ambient temperature. The excitation wavelength was 320 nm. The emission spectra were recorded with a scan speed of 200 nm min^−1^, in the domain 350–700 nm, with a 350 nm cut-off filter.

The X-ray diffractograms (XRD) were obtained with PANalytical Empyrean equipment (from Malvern PANalytical, Bruno, Nederland) using λCuKα = 1.54184 Å.

A high-resolution transmission electron microscope, Tecnai G2F30 S-TWIN (FEI Company, Eindhoven, The Netherlands) was used to acquire the transmission electron (TEM) images for ZnO nanoparticles. The scanning electron microscopy (SEM) was performed on a Tescan VEGA 3 LM (Tescan, Brno, Czech Republic).

For the photocatalytic activity determination, a solution obtained by mixing 100 mL of stock solution for each dye was used. The stock solutions were prepared with concentrations of 10 mg/L methylene blue (MB), 20 mg/L methyl orange (MO), and 12 mg/L rhodamine B (RhB). For each test, a 20 mg sample ZnO powder was added over 10 mL mixed dye solution and stirred for 30 min in the dark to reach adsorption–desorption equilibrium. The samples were irradiated with a visible light fluorescent lamp of 160 W/2900 lm LOHUIS^®^ (Lohuis, Bucharest, Romania), placed at a 0.1 m distance. The reactor setup is similar to the one depicted in [77]. A sample was extracted at defined time intervals and the concentration of each dye was measured with UV-Vis spectrophotometer at the specific maximum of each dye.

Determination of the antibacterial activity was made against seven reference microorganisms by American Type Culture Collection (ATCC, Manassas, VA, USA): three Gram-negative bacterial strains *Escherichia coli* (*E. coli*) ATCC 25922; *Salmonella enterica* serovar Typhimurium (*S. typhimurium*) ATCC 14028; *Pseudomonas aeruginosa* (*P. aeruginosa*) ATCC 27853; and four Gram-positive bacterial strains *Staphylococcus aureus* (*S. aureus*) ATCC 25923; *Bacillus cereus* (*B. cereus*) ATCC 19659; *Bacillus subtilis* (*B. subtilis*) ATCC 6633, and *Enterococcus faecalis* (*E. faecalis*) ATCC 29212. In order to determine the antimicrobial activity of ZnO nanoparticles, we employed an adapted diffusion assay, presented in [78] and respecting the general rules exposed in the CLSI 2020. A 0.5 McFarland bacterial suspension (1.5 × 10^8^ CFU/mL), previously prepared in sterile saline (0.9% NaCl solution), was utilized as a standardized inoculum to swab inoculate Petri dishes containing nutritive agar. A 10 mg/mL ZnO suspension was prepared by using Milli-Q water as medium. Homogeneity of the suspension was obtained by sonication for 30 min. Drops with a volume of 10 μL from the ZnO suspension were added to the inoculated Petri dishes and these were incubated for 20 h at 37 °C. After incubation, the diameter of growth inhibition developed around each material specimen was measured and expressed in mm.

Planktonic growth in the presence of the obtained ZnO nanoparticles was analyzed in nutritive broth. Then, 10 μL ZnO suspension were placed in sterile 24-well plates. Then, 1 mL of nutritive broth and 10 μL of the previously obtained 0.5 McFarland bacterial suspensions in PBS were added. Specimens were allowed to incubate for 24 h at 37 °C. To spectrophotometrically evaluate the growth of planktonic (free-floating) cultures, 0.15 mL of the obtained bacterial culture were transferred to 96-well plates and the absorbance at 600 nm was evaluated.

The obtained results were statistically analyzed using the analysis of variance (ANOVA) performed with Microsoft Excel 2016 (Microsoft Corp., Redmond, WA, USA), having installed the XLSTAT 2020.5.1 add-on. The Shapiro–Wilk test was used to check the normal distribution of the data. By Levene’s test, we assessed the homoscedasticity of the residuals. The results were compared by Tukey’s (HSD) test so that the pairs of films that differed in terms of statistical significance were revealed (where *p* < 0.05).

## 4. Conclusions

In this study, we have obtained ZnO NPs by forced solvolysis, in three different alcohols: n-butanol, ethylene glycol and glycerin. The last synthesis leads to zinc glycerolate, which must be further refluxed in water to obtain ZnO NPs. This indicates that, in principle, only the alcohols with one or two hydroxyl moieties are suitable to obtain ZnO nanoparticles in a single step synthesis. The ZnO NPs have different sizes and morphologies, depending on the alcohol used, the alcohols with single –OH group being most suitable for obtaining small nanoparticles. The ZnO NPs obtained in n-butanol are monocrystalline, polyhedral shaped with an average size of 27 nm. The nanoparticles obtained in ethylene glycol are also monocrystalline, round shaped, with a size of ~ 44 nm. Finally, the particles obtained from glycerin are larger, ~120 nm, hexagonal, and polycrystalline.

The samples were tested for possible uses in water purification applications, as photo catalysts under visible light and disinfectants. The photocatalytic activity was tested against a mixture of three dyes belonging to different dye classes, methylene blue (MB), rhodamine B (RhB), and methyl orange (MO). For all the samples, the photocatalytic degradation efficiency was in the order MO > RhB > MB, with differences increasing with the size of the nanoparticles. The ZnO NPs with smallest size exhibited a remarkable degradation efficiency of over 99%, maintaining solid >90% efficiency over five photocatalytic cycles. At the same time, the sample with the smallest size of nanoparticles exhibited the strongest disinfection potential against a broad range of bacterial strains, both Gram-positive and -negative. These results permit the further use of ZnO nanoparticles in technical and medical disinfection applications.

## Figures and Tables

**Figure 1 ijms-24-05677-f001:**
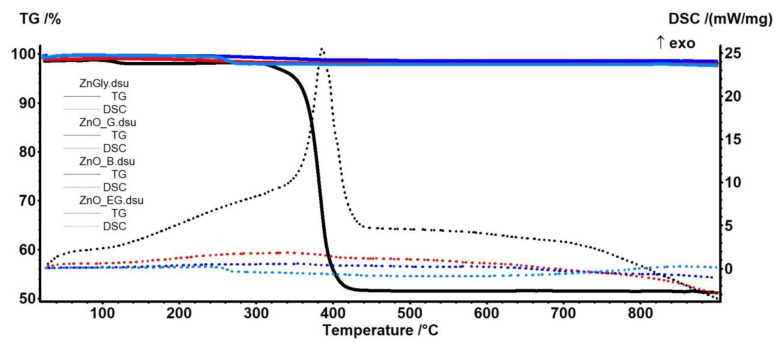
The thermogravimetry (solid) and differential scanning calorimetry (dotted) curves for the ZnO_B, ZnO_EG, ZnO_G and ZnGly samples.

**Figure 2 ijms-24-05677-f002:**
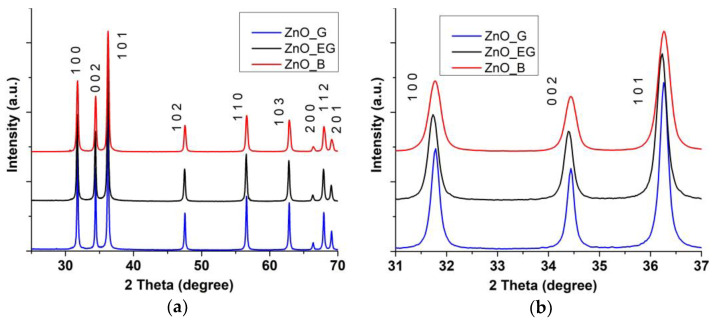
The X-ray diffractograms for: (**a**) ZnO_B, ZnO_EG and ZnO_G samples; (**b**) detail for the 2θ = 31–37° interval.

**Figure 3 ijms-24-05677-f003:**
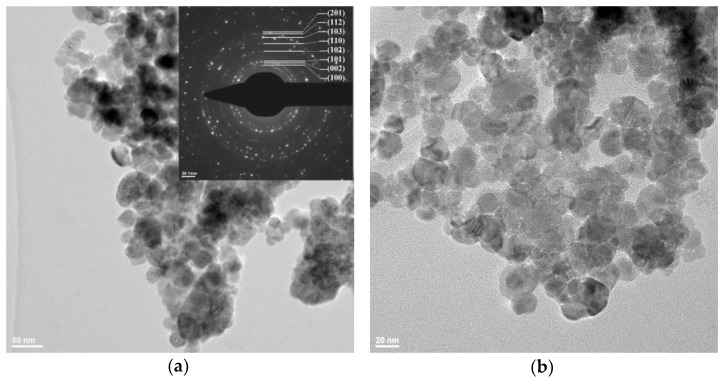
The TEM images for ZnO_B sample (**a**) 39,000× magnification and SAED pattern of hexagonal ZnO (JCPDS card no. 80-0075) in inset; (**b**,**c**) 75,000× magnification; (**d**) high resolution TEM with (0 0 2) crystallographic planes.

**Figure 4 ijms-24-05677-f004:**
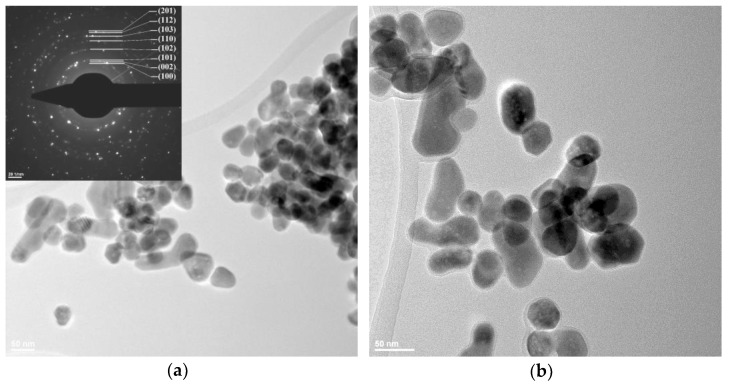
The TEM images for ZnO_EG samples (**a**) 29,500× magnification and SAED pattern of hexagonal ZnO (JCPDS card no. 80-0075) in inset; (**b**) 49,000× magnification; (**c**) 75,000× magnification; (**d**) high resolution TEM with (0 0 2) crystallographic planes.

**Figure 5 ijms-24-05677-f005:**
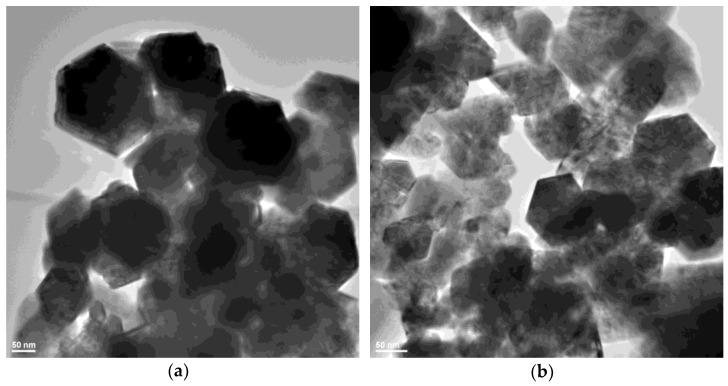
The TEM images for ZnO_G samples (**a**) 29,500× magnification; (**b**) 39,000× magnification; (**c**) 49,000× magnification; (**d**) high resolution TEM and identified planes of hexagonal ZnO (JCPDS card no. 80-0075).

**Figure 6 ijms-24-05677-f006:**
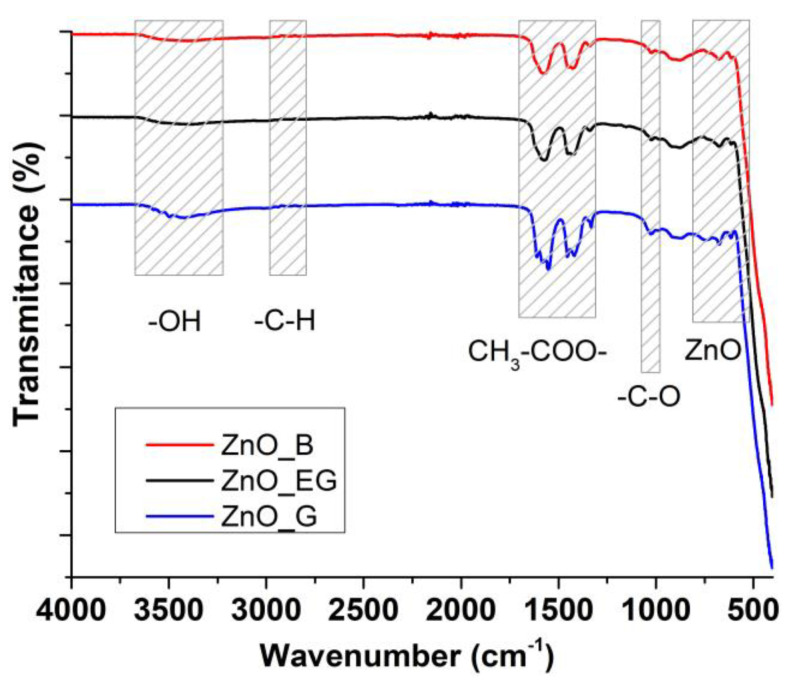
The FTIR spectra for ZnO_B, ZnO_EG and ZnO_G samples.

**Figure 7 ijms-24-05677-f007:**
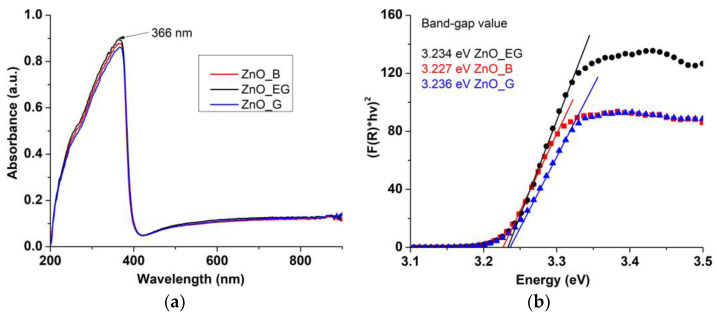
The UV-Vis spectra for ZnO_B, ZnO_EG and ZnO_G samples (**a**); the band-gap energy values (**b**).

**Figure 8 ijms-24-05677-f008:**
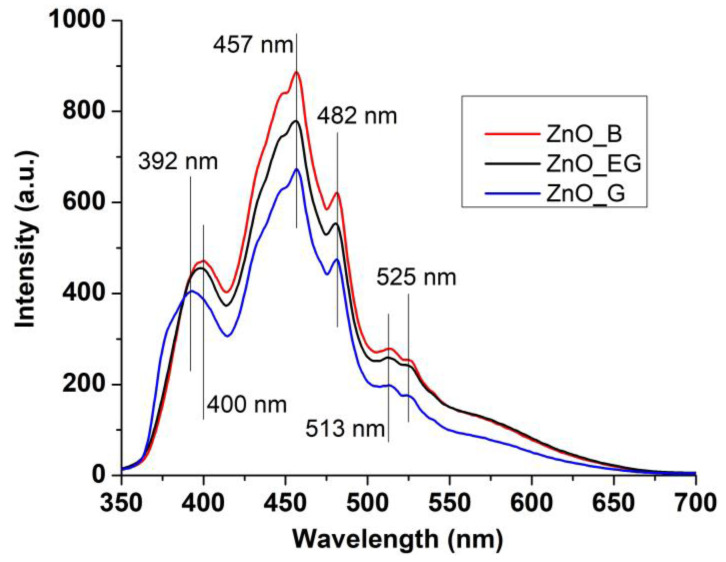
The emission spectra for ZnO_B, ZnO_EG and ZnO_G samples.

**Figure 9 ijms-24-05677-f009:**
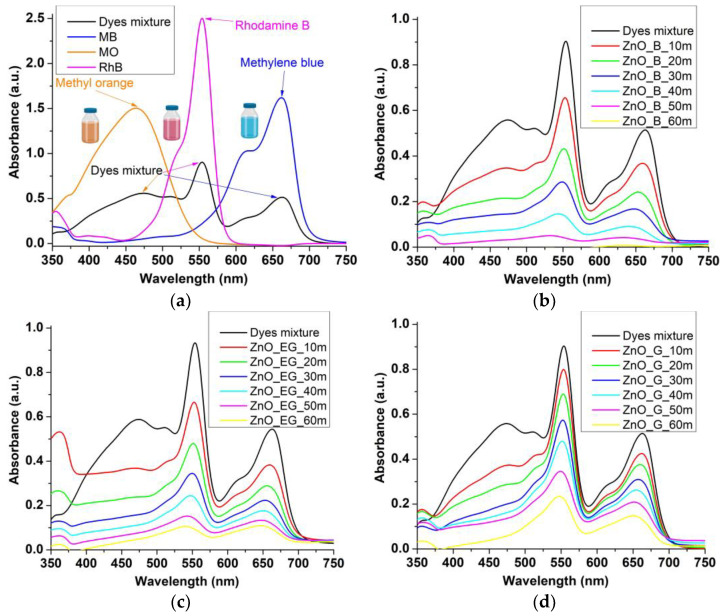
Absorption spectra for individual dyes, methylene blue (MB), Rhodamine B (RhB) and methyl orange (MO) and for dyes mixture (MB + RhB + MO), with indication of the corresponding absorption maxima (**a**); photocatalytic activity of ZnO_B sample against dyes mixture (**b**); photocatalytic activity of ZnO_EG sample against dyes mixture (**c**); photocatalytic activity of ZnO_G sample against dyes mixture (**d**).

**Figure 10 ijms-24-05677-f010:**
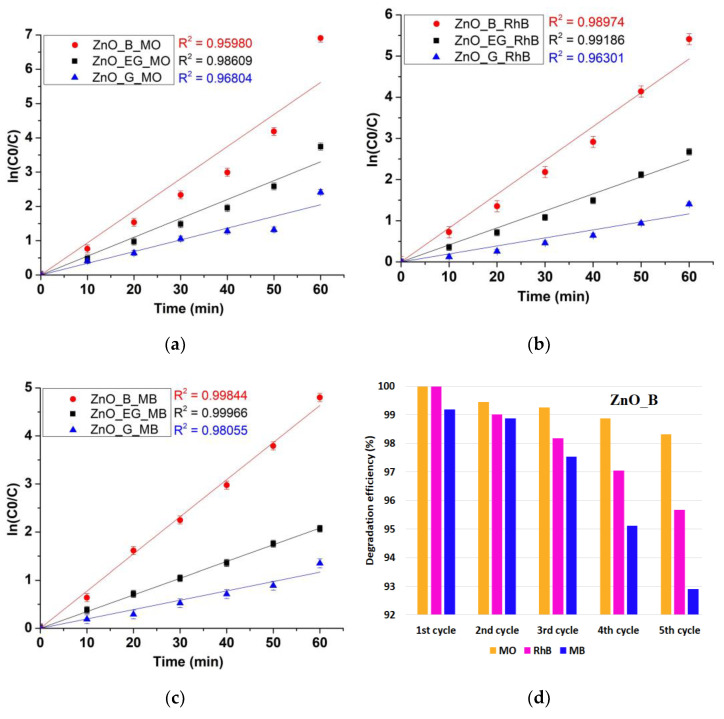
Determination of *k*_app_ by plotting ln(C_0_/C) vs time for each dye from the MO + RhB + MB mixture: MO (**a**); RhB (**b**); MB (**c**); degradation efficiency for ZnO_B sample over five photocatalytic cycles against MO + RhB + MB mixture (**d**).

**Figure 11 ijms-24-05677-f011:**
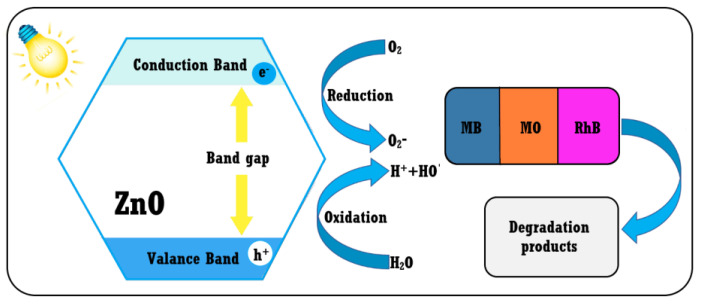
Photocatalytic activity against MB, MO, EY, GV and RhB for ZnO_C4 sample.

**Figure 12 ijms-24-05677-f012:**
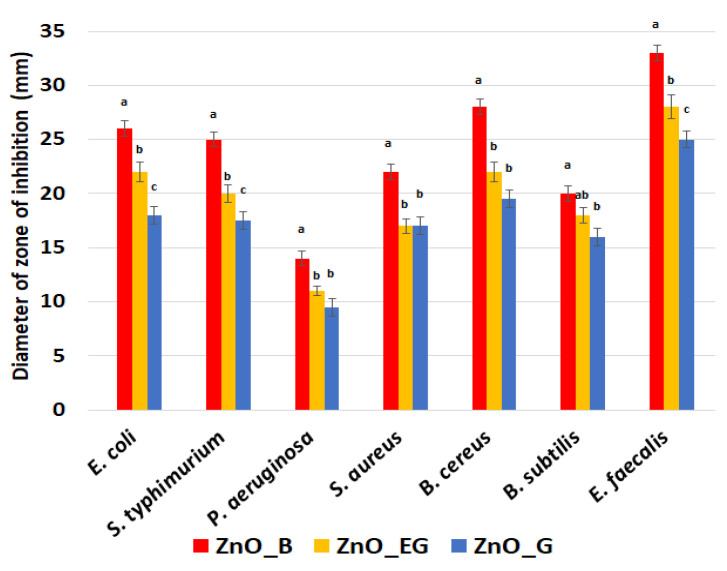
Growth inhibition diameters (mm) measured for each bacterial strains in the presence of ZnO NPs. Different small letters indicates statistically significant differences between films (*p* < 0.05).

**Figure 13 ijms-24-05677-f013:**
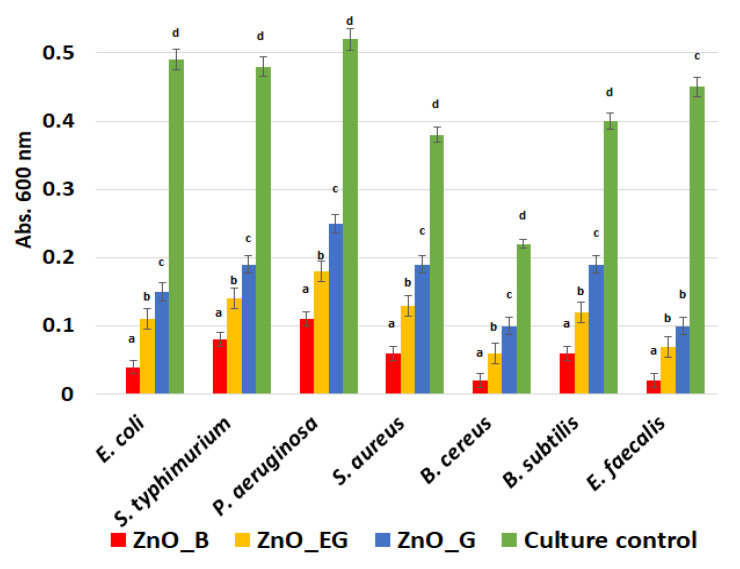
Graphic representation of average absorbance at 600 nm revealing growth of planktonic bacterial cultures in the presence of control and ZnO NPs for 24 h at 37 °C. Different small letters (a–d) indicate statistically significant differences between films (*p* < 0.05).

**Figure 14 ijms-24-05677-f014:**
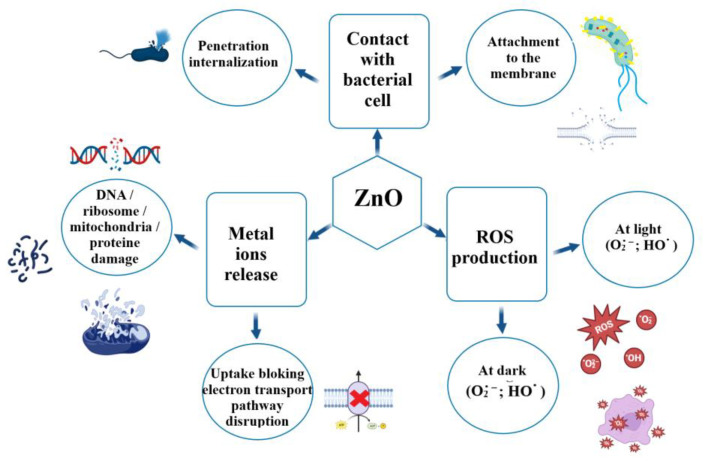
Graphic representation of proposed antibacterial mechanisms for ZnO nanoparticles.

**Table 1 ijms-24-05677-t001:** The principal data from thermal analysis.

Sample	Mass Loss (%) 20–200 °C	Mass Loss (%) 200–900 °C	Endothermal/Exothermal	Residual Mass (%)
ZnO_B	0.41%	1.12%	66.2/357.8 °C	98.47%
ZnO_EG	0.47%	1.92%	113.7/275.8 °C	97.60%
ZnO_G	1.19%	0.72%	71.9/341.5 °C	98.09%

**Table 2 ijms-24-05677-t002:** The calculated lattice parameters for the ZnO_B, ZnO_EG and ZnO_G samples.

Sample	ZnO_B	ZnO_EG	ZnO_G
**Unit cell**			
***a* = *b* [Å]**	3.25076	3.25097	3.25073
***c* [Å]**	5.20665	5.20742	5.20685
**V [Å^3^]**	47.64955	47.66275	47.65050
** *c/a* **	1.60167	1.60181	1.60175
**Microstrain (%)**	0.26 ± 0.11	0.20 ± 0.05	0.25 ± 0.07
**Crystallite size (D)**	31.51 ± 2.47	45.14 ± 2.59	42.13 ± 3.61
**Dislocation density (δ) × 10^−4^**	10.07	4.91	5.63

**Table 3 ijms-24-05677-t003:** The determined band-gap energy values for ZnO nanopowders.

Sample	ZnO_B	ZnO_EG	ZnO_G
**Band gap value (eV)**	3.227	3.234	3.236

**Table 4 ijms-24-05677-t004:** Calculated photocatalytic parameters: degradation efficiency, *k*_app_ and R^2^ for ZnO_B, ZnO_EG and ZnO_G samples against dyes mixture solution.

**Sample** **Label**	**Degradation Efficiency** **(%)**			***k*_app_ × 10^3^** **(min^−1^)**			**R^2^**		
	MO	RhB	MB	MO	RhB	MB	MO	RhB	MB
**ZnO_B**	99.99	99.99	99.18	93.62 ± 7.22	82.19 ± 3.16	77.24 ± 1.15	0.95980	0.98974	0.99844
**ZnO_EG**	97.63	93.11	87.38	55.05 ± 2.47	41.34 ± 1.41	34.69 ± 0.24	0.98609	0.99186	0.99966
**ZnO_G**	91.07	75.43	74.06	34.17 ± 2.34	19.46 ± 1.44	19.47 ± 1.04	0.96804	0.96301	0.98055

**Table 5 ijms-24-05677-t005:** Sample labels for each solvent used.

Sample Code	Alcohol	Obtained Product
ZnO_B	n-Butanol	ZnO
ZnO_EG	Ethylene glycol	ZnO
ZnGly	Glycerin	Zinc glycerolate
ZnO_G	Glycerin/water	ZnO

## Data Availability

Not applicable.

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
