# Peer review of "Antibacterial Activity of Solvothermal Obtained ZnO Nanoparticles with Different Morphology and Photocatalytic Activity against a Dye Mixture: Methylene Blue, Rhodamine B and Methyl Orange"

_ijms, 2023, doi:10.3390/ijms24065677_

Round 1
Reviewer 1 Report
I have received the following paper, submitted to International Journal of Molecular Sciences entitled: Antibacterial activity of solvothermal obtained ZnO nanoparticles with different morphology and photocatalytic activity against a dye mixture: methylene blue, rhodamine B and methyl orange.
The study raises some critical issues concerning the synthesis and application of zinc oxide nanoparticles in the removal of dyes and the purification of water from microbial pollutants, particularly bacteria. However, some points need to be clarified or changed to the paper becomes peer-reviewed.
1- The introductory sentences should be reduced from the abstract and replaced with a presentation of methods and results.
2- The introduction should be rewritten because it clearly lacks the hypothesis and aim. It also has some repeated sentences in the results and discussion that should be removed. The final section (lines 98-105) should be moved to the conclusion.
3- The authors state in lines 66 and 408 of the results that Zn+2 ions can be liberated from ZnO NPs and affect bacteria. Is this something you can get? And, if the situation is as you stated, I believe zinc ions in any type of zinc salt will be a cheaper and more effective alternative to zinc nanoparticles. It requires clarification.
4- Lines 134-135, the authors state that they performed SEM even though it is not included in the method!!! How does that sound?
5- How do your account for the large differences in the size of nano-zinc particles measured using TEM, XRD, and DLS, as shown in the provided supplementary data?
6- In the antibacterial activity assay, the authors did not mention how the nanoparticle suspension was prepared, nor did they use any control solvent to indicate it in the results. Because ZnO NPs do not dissolve in water, they are not bioavailable. What are your thoughts on this issue?
7- Although they used the CLSI guideline as their point of reference, the authors did not specify the type of media employed or any type of control in the antibacterial activity test carried out utilizing the diffusion method. To assess the applicability of the methods employed in the experiment, antibiotic control was to be utilized. It was also preferable to create a second control using zinc acetate at an equivalent concentration in ZnO NPs to compare the two. Is that even a possibility?
8- Based on the guidelines for this method, the OD values in the two samples of ZnO-G and ZnO-EG in the micro-dilution experiment were greater than 0.1, and this indicates that all the bacteria used in the test, except E. feacalis, were able to overcome these substances at the concentration used. The results in the aforementioned situations only show a partial inhibition of bacterial growth rather than a whole eradication, as was the case with the use of ZnO-B. (except for P. aeruginosa bacteria). Therefore, it was suggested to measure the minimum inhibitory concentration of these nanoparticles on the mentioned bacterial species.
9- In line 460, in the photocatalytic activity test, why were different concentrations of dyes used to make the mixture and not an equal proportion of them?
10- In line No. 462, don't you see that the concentration of nanoparticles used to remove dyes is large and may be unacceptable in application because it could pollute the environment from the other side?
11- In the method, the photocatalytic activity determination test needs more details and more description.
12- The conclusion should extract cognitive information from the data presented in the results and discussion sections rather than repeating it, please rephrase it.
Author Response
I have received the following paper, submitted to International Journal of Molecular Sciences entitled: Antibacterial activity of solvothermal obtained ZnO nanoparticles with different morphology and photocatalytic activity against a dye mixture: methylene blue, rhodamine B and methyl orange.
The study raises some critical issues concerning the synthesis and application of zinc oxide nanoparticles in the removal of dyes and the purification of water from microbial pollutants, particularly bacteria. However, some points need to be clarified or changed to the paper becomes peer-reviewed.
R: We are grateful to the esteem reviewer for the appreciation words. Following the helpful advices received, we further improved the manuscript and corrected the indicated mistakes. We hope that the esteem reviewer will find it suitable for publishing.
1- The introductory sentences should be reduced from the abstract and replaced with a presentation of methods and results.
R1: We have reduced the size of the abstract by removing the general introductory sentences, and maintained only the short description of methods and obtained results.
2- The introduction should be rewritten because it clearly lacks the hypothesis and aim. It also has some repeated sentences in the results and discussion that should be removed. The final section (lines 98-105) should be moved to the conclusion.
R2: We have trim the introduction and final lines were moved to conclusion section. We have added new paragraphs, with clear indication of hypothesis and aim of the study.
3- The authors state in lines 66 and 408 of the results that Zn+2 ions can be liberated from ZnO NPs and affect bacteria. Is this something you can get? And, if the situation is as you stated, I believe zinc ions in any type of zinc salt will be a cheaper and more effective alternative to zinc nanoparticles. It requires clarification.
R3: According to the proposed mechanism for the antibacterial mechanism, and according to the mentioned literature, one of the pathways is represented by the release of Zn2+ ions inside the cellular cytoplasm, after the nanoparticles internalization (just one of the possible mechanisms). The Zn2+ ions can coordinate various components and disrupt the normal function of organelle. Adding straight a zinc salt might not be an efficient way as zinc ions have to pass the cellular membrane first.
In addition, only ZnO has photocatalytic activity, not zinc salts. Zinc salts would dissolve in water and would not be possible to remove them by simple decantation at the end of water treatment. And this study indicates that ZnO NPs can be used in water treatment, as an antibacterial agent, but also as a way to photo-decompose various organic pollutants.
4- Lines 134-135, the authors state that they performed SEM even though it is not included in the method!!! How does that sound?
R4: We are very grateful to the esteem reviewer for pointing out this weakness. Description of all the instruments and methods employed are presented in section “3. Materials and Methods”. We have added the missing information about SEM. The supplementary material contains the SEM micrographs as we feel they were only marginally related with the study.
5- How do your account for the large differences in the size of nano-zinc particles measured using TEM, XRD, and DLS, as shown in the provided supplementary data?
R5: The resulting size of particles can be different as function of the used method. In a TEM image some particles can be seen and measured. By using software ImageJ we measured hundreds particles in order to obtain a closer to reality average. In XRD only the crystallite size can be measured. Some samples are monocrystalline, e.g. each particle is formed from a single grain, or they can be polycrystalline, e.g. in a particle there are many fused crystallite (like in sample ZnO_G). We did not perform DLS measurements, but in such case the hydrodynamic diameter is obtained, e.g. the particle + closer water molecule shell, the results being always larger than those obtained in TEM for example.
6- In the antibacterial activity assay, the authors did not mention how the nanoparticle suspension was prepared, nor did they use any control solvent to indicate it in the results. Because ZnO NPs do not dissolve in water, they are not bioavailable. What are your thoughts on this issue?
R6: We have used ultrapure water for the ZnO suspension, the homogeneity being obtained with a ultrasound bath. Relevant information was inserted in the manuscript. It is the same method used for all inorganic nanoparticles that are insoluble in water (ZnO, TiO2, Fe3O4, Ag, SiO2 etc).
7- Although they used the CLSI guideline as their point of reference, the authors did not specify the type of media employed or any type of control in the antibacterial activity test carried out utilizing the diffusion method. To assess the applicability of the methods employed in the experiment, antibiotic control was to be utilized. It was also preferable to create a second control using zinc acetate at an equivalent concentration in ZnO NPs to compare the two. Is that even a possibility?
R7: The antimicrobial activity of the samples was evaluated by the diffusion assay. The diffusion method is suitable for identifying the most active antimicrobial agents but not for quantifying bioactivity. This method is based on the determination of the zone of inhibition proportional to the microbial susceptibility to the presence of ZnO. The zone of inhibition defines the extent of antibacterial activity and is expressed as the diameter of this zone in mm. The size of this zone depends on the rate of diffusion and cell growth.
8- Based on the guidelines for this method, the OD values in the two samples of ZnO-G and ZnO-EG in the micro-dilution experiment were greater than 0.1, and this indicates that all the bacteria used in the test, except E. feacalis, were able to overcome these substances at the concentration used. The results in the aforementioned situations only show a partial inhibition of bacterial growth rather than a whole eradication, as was the case with the use of ZnO-B. (except for P. aeruginosa bacteria). Therefore, it was suggested to measure the minimum inhibitory concentration of these nanoparticles on the mentioned bacterial species.
R8: We are very grateful to the esteem reviewer for pointing out this weakness. The samples have different size and morphology. The smallest nanoparticles are in case of ZnO_B <30 nm and the largest are in case of ZnO_G ~ 120 nm. The ZnO-B sample has polyhedral monocrystalline nanoparticles, ZnO_EG has rounded monocrystalline nanoparticles and ZnO_G has hexagonal polycrystalline particles. The best antibacterial activity was obtained for the sample with the smallest nanoparticle size. Information pointed out by the esteem reviewer was introduced in the article. Unfortunately, we do not have the opportunity to measure also MIC as the focus of this article is on photocatalytic activity against a dye mixture (which is also the novelty of the study, as antibacterial activity of ZnO is already reported in literature).
9- In line 460, in the photocatalytic activity test, why were different concentrations of dyes used to make the mixture and not an equal proportion of them?
R9: We thank the esteem reviewer for giving us the opportunity to explain this part. We have used solutions with concentrations of dyes that can be found in other literature reports. As dyes have different molar extinction coefficients, their absorbance would be way different if used in same concentration (e.g. 10 mg /L MB exhibit ~ same absorption intensity as 20 mg/L MO). Also, as we rely on Lambert-Beer law, to consider absorbance as a first order function of concentration, the absorbance should not exceed two units (better under one unit) in case of Jasco V560.
10- In line No. 462, don't you see that the concentration of nanoparticles used to remove dyes is large and may be unacceptable in application because it could pollute the environment from the other side?
R10: The quantity of photocatalyst that is used in these experiments is the usual one reported in other literature studies, between 1-2 mg ZnO/ 1 mL solution (doi: 10.1080/26395940.2022.2081261; doi 10.1155/2020/1768371; doi: 10.1021/acsomega.0c05092; doi: 10.3390/molecules27010006; doi: 10.2166/wst.2021.360; doi: 10.1007/s11051-022-05421-7 etc. Scaling up the process from laboratory to industrial size is subject to further optimization processes and flux. A decantation stage must be employed, regardless of the quantity of photocatalyst used, as a method to recycle it.
11- In the method, the photocatalytic activity determination test needs more details and more description.
R11: We have expanded the description of the method with a reference (doi: 10.1155/2020/1768371) where a similar reactor is depicted. We feel that is not necessarily to insert in the section 3 an additional figure, beside the description made. The solution used in photocatalytic test is obtained by mixing stock solutions of MB, MO and RhB, like depicted in figure 9a. The used lamp type and manufacturer are presented, as well as the distance from the irradiated samples. The concentration of each dye was measured with UV-Vis spectrophotometer at each dye specific maximum.
12- The conclusion should extract cognitive information from the data presented in the results and discussion sections rather than repeating it, please rephrase it.
R12: We wish to thank to the esteem reviewer for indicating how we can improve the manuscript. We have rephrased some of the Conclusion section indicating the most important information.
Reviewer 2 Report
Motelica et al. reported different morphologies of ZnO NPs and their photocatalytic antibacterial activity for clean water. There are some concerns listed below
1. Explain the mechanism for bacteria too and these are some examples that can be used to refer https://www.ncbi.nlm.nih.gov/pmc/articles/PMC8405884/, https://www.mdpi.com/2079-4991/10/4/643/htm
2. please draw a graphical abstract
3. ZnO NPs are already established antimicrobial substances how do you defend the novelty of this research?
4. try reducing number of references in introduction.
5. Graphpad prism or origin should be used to plot the graph and statistical significance especially Figure 12& 13.
Author Response
Motelica et al. reported different morphologies of ZnO NPs and their photocatalytic antibacterial activity for clean water. There are some concerns listed below
We are grateful to the esteem reviewer for the time and effort spent to help us improve the manuscript. Following the helpful advices received, we further improved the manuscript and we hope that the esteem reviewer will find it suitable for publishing.
- Explain the mechanism for bacteria too and these are some examples that can be used to refer https://www.ncbi.nlm.nih.gov/pmc/articles/PMC8405884/, https://www.mdpi.com/2079-4991/10/4/643/htm
R1: We thank esteem reviewer for indicating this valuable studies. We have proposed a mechanism for the antibacterial activity and acknowledge them.
- please draw a graphical abstract
R2: We are very grateful to the esteem reviewer for pointing out this weakness. A graphical abstract was loaded in the system.
- ZnO NPs are already established antimicrobial substances how do you defend the novelty of this research?
R3: While antimicrobial activity of ZnO NPs is known, the exact mechanism is still open to debate. The antimicrobial activity is function of size, shape and surface defects of the particles. Here we report synthesis of ZnO NPs by forced solvolysis, without need of NaOH or thermal calcination. The alcohol type used in synthesis has influenced the types of particles that were obtained, hence the antimicrobial activity is different. In addition, the application presented here is related to the photocatalytic activity against three different dyes, simultaneous in a mixture, which is rarely reported in literature, despite water being contaminated with a mixture of pollutants in reality. This is very different from the photocatalytic study made on each individual dye, as competition between dyes can occur.
- try reducing number of references in introduction.
R4: We have polished the introduction section with clear hypothesis and aim. The reference number was halved.
- Graphpad prism or origin should be used to plot the graph and statistical significance especially Figure 12& 13.
R5: We are thankful to the esteem reviewer for the suggestion. We have used XLSTAT add-on for Excel in order to plot the graphs and statistical significance, which is also specialized for statistical data processing.
Round 2
Reviewer 1 Report
Although the authors made some effort to clarify some of the previous questions and inquiries, they did not provide a clear response to others, particularly comments 3 to 8. Consequently, I believe that the authors should conduct the research in order for it to be peer-reviewed.
Author Response
Although the authors made some effort to clarify some of the previous questions and inquiries, they did not provide a clear response to others, particularly comments 3 to 8. Consequently, I believe that the authors should conduct the research in order for it to be peer-reviewed.
We are grateful to the esteem reviewer for the time and effort spent to help us improve the manuscript. Following the helpful advices received, we are detailing our previous responses at points 3 to 8, with literature references, as we feel that such detailed responses will clarify the issues raised:
3- The authors state in lines 66 and 408 of the results that Zn+2 ions can be liberated from ZnO NPs and affect bacteria. Is this something you can get? And, if the situation is as you stated, I believe zinc ions in any type of zinc salt will be a cheaper and more effective alternative to zinc nanoparticles. It requires clarification.
R3: The proposed application for this ZnO nanoparticles is water treatment under visible light, that will permit antibacterial action but also removal of organic pollutants! Zinc oxide has antimicrobial activity, and literature reports can be consulted (doi: 10.1016/j.sdentj.2018.06.003; doi: 10.2147/IJN.S216204; doi: 10.1093/rb/rbac019; doi: 10.1039/C9TB02258A; doi: 10.1186/s11671-018-2532-3). The main issue is that any zinc salts cannot be used for water depollution as they are soluble and will not exhibit photocatalytic activity. Zinc salts will kill microorganisms but will not decompose the harmful organic pollutants. One further problem is that such salts should be removed from water after treatment, and this is an expensive process. On the other hand, ZnO nanoparticles will simply sediment and therefore they can be easily removed from water. ZnO nanoparticles have both: antimicrobial and photocatalytic activity! The discussion in the article is about the mechanisms involved in the antimicrobial activity of ZnO. One of the mechanisms is the release of some zinc ions near or inside microorganisms cell, and the proposed mechanisms are in accordance with literature reports, doi: 10.1038/s41598-022-06657-y; doi: 10.1590/0104-6632.20190362s20180027; doi: 10.3390/polym15030529; doi: 10.3389/fphy.2021.641481; doi: 10.1007/s40820-015-0040-x. The other antibacterial mechanisms involve generation of ROS and mechanical damage of cellular membrane (doi: 10.1016/j.serj.2017.10.001)
Therefore, we feel that our answer is complete to this issue. Yes, zinc ions do have antimicrobial action, but, no, zinc salts cannot be used in the same way as ZnO nanoparticles in water treatment.
4- Lines 134-135, the authors state that they performed SEM even though it is not included in the method!!! How does that sound?
R4: As we previously mentioned, we are very grateful to the esteem reviewer for pointing out this oversight. Description of all the instruments and methods employed are presented now in section “3. Materials and Methods”. We have added the missing information about SEM: “The scanning electron microscopy (SEM) was performed on a Tescan VEGA 3 LM (Tescan, Brno, Czech Republic)” rows 487-488.
5- How do your account for the large differences in the size of nano-zinc particles measured using TEM, XRD, and DLS, as shown in the provided supplementary data?
R5: At this issue we have previously replied, explaining why some differences can appear between sizes determined in TEM/SEM or in XRD. We did not perform any DLS experiments, nevertheless we included the differences that are induced by this technique also. The resulting size of particles can be different as function of the used method. In a TEM micrograph the particles are measurable vs the scale bar with software like ImageJ. In XRD only the crystallite size can be evaluated by using Scherrer equation for example. A nanoparticle can have one or more crystallite in it. TEM/SEM and XRD sizes are in good match for monocrystalline samples, while for samples that contain more than one grain, the TEM/SEM size will always be larger than XRD determined size. We did not perform DLS measurements, but in such case the hydrodynamic diameter is obtained, e.g. the particle + closer water molecule shell, the results being always larger than those obtained in TEM for example. (doi: 10.1007/s11051-012-1269-7; doi: 10.1021/la0477183; doi: 10.1021/es0705543; doi: 10.1016/j.cplett.2011.11.049)
6- In the antibacterial activity assay, the authors did not mention how the nanoparticle suspension was prepared, nor did they use any control solvent to indicate it in the results. Because ZnO NPs do not dissolve in water, they are not bioavailable. What are your thoughts on this issue?
R6: We are grateful for pointing out this issue. We previously indicated that we have used ultrapure water for the ZnO suspension, the homogeneity being obtained with an ultrasound bath. Relevant information was inserted in the manuscript (row 510). It is the same method used for all inorganic nanoparticles that are insoluble in water (ZnO, TiO2, Fe3O4, Ag, SiO2 etc). Such inorganic nanoparticles present antimicrobial activity, but they are insoluble, therefore always for testing purposes a suspension must be prepared. Other solvents can influence the antimicrobial results, but water has no antibacterial activity (doi: 10.1371/journal.pone.0259190).
7- Although they used the CLSI guideline as their point of reference, the authors did not specify the type of media employed or any type of control in the antibacterial activity test carried out utilizing the diffusion method. To assess the applicability of the methods employed in the experiment, antibiotic control was to be utilized. It was also preferable to create a second control using zinc acetate at an equivalent concentration in ZnO NPs to compare the two. Is that even a possibility?
R7: The antimicrobial activity of the samples was evaluated by the adapted diffusion assay. The diffusion method is suitable for identifying the most active antimicrobial agents but not for quantifying bioactivity. In this case the antibiotic controls are not necessary, since all the tested strains are ATCC (laboratory strains with known antibiotic susceptibility, as revealed by the CLSI standard). Therefore, the diameter of inhibition zone, obtained for the evaluated NPs samples could be compared with the results of the QC (quality control) found in the CLSI standard for each traditional antibiotic.
8- Based on the guidelines for this method, the OD values in the two samples of ZnO-G and ZnO-EG in the micro-dilution experiment were greater than 0.1, and this indicates that all the bacteria used in the test, except E. feacalis, were able to overcome these substances at the concentration used. The results in the aforementioned situations only show a partial inhibition of bacterial growth rather than a whole eradication, as was the case with the use of ZnO-B. (except for P. aeruginosa bacteria). Therefore, it was suggested to measure the minimum inhibitory concentration of these nanoparticles on the mentioned bacterial species.
R8: As we previously mentioned, we are very grateful to the esteem reviewer for pointing out this weakness. The samples have different size and morphology. The smallest nanoparticles are in case of ZnO_B <30 nm and the largest are in case of ZnO_G ~ 120 nm. The ZnO-B sample has polyhedral monocrystalline nanoparticles, ZnO_EG has rounded monocrystalline nanoparticles and ZnO_G has hexagonal polycrystalline particles. The best antibacterial activity was obtained for the sample with the smallest nanoparticle size. Information pointed out by the esteem reviewer was introduced in the article (rows 415-417). The other information is already present in section 3.
Indeed, we did not presented MIC as it was not determined. The literature reported MIC and MBC values are in the range of micrograms/mL, while we use for photocatalytic depollution concentration of milligrams/mL, so three order higher. Regardless of exact MIC value, the application proposed (water treatment) will benefit of ZnO concentrations many times higher, that will generate automatically the antibacterial activity (doi: 10.3390/molecules27113532; doi: 10.1016/j.jksus.2022.102110; doi: 10.14202/vetworld.2018.1428-1432; doi: 10.1016/j.saa.2012.01.006; doi: 10.2147/IDR.S221408)